# Mannich Base Derived from Lawsone Inhibits PKM2 and Induces Neoplastic Cell Death

**DOI:** 10.3390/biomedicines12122916

**Published:** 2024-12-21

**Authors:** Lucas Rubini-Dias, Tácio V. A. Fernandes, Michele P. de Souza, Déborah Hottz, Afonso T. Arruda, Amanda de A. Borges, Gabriel Ouverney, Fernando de C. da Silva, Luana da S. M. Forezi, Gabriel Limaverde-Sousa, Bruno K. Robbs

**Affiliations:** 1Programa de Pós-Graduação em Ciências Morfológicas, Instituto de Ciências Biomédicas, Universidade Federal do Rio de Janeiro, Fundão, Rio de Janeiro 21941-590, RJ, Brazil; lrubini@ufrj.br (L.R.-D.); afonsothales@id.uff.br (A.T.A.); ouverneygabriel@ufrj.br (G.O.); 2Departamento de Síntese de Fármacos, Instituto de Tecnologia em Fármacos, Farmanguinhos–Fiocruz, Manguinhos, Rio de Janeiro 21041-250, RJ, Brazil; tacio.fernandes@fiocruz.br; 3Postgraduate Program in Applied Science for Health Products, Faculty of Pharmacy, Fluminense Federal University, Niterói 24020-141, RJ, Brazil; michelep410@gmail.com; 4Departamento de Ciência Básica, Instituto de Saúde de Nova Fribrugo, Universidade Federal Fluminense, Nova Friburgo 28625-650, RJ, Brazil; deborah.hottz@gmail.com; 5Departamento de Química Orgânica, Instituto de Química, Campus do Valonguinho, Universidade Federal Fluminense, Niterói 24020-150, RJ, Brazil; borgesamanda@id.uff.br (A.d.A.B.); fcsilva@id.uff.br (F.d.C.d.S.); luanaforezi@id.uff.br (L.d.S.M.F.); 6Instituto Oswaldo Cruz, Fundação Oswaldo Cruz, Rio de Janeiro 21040-900, RJ, Brazil

**Keywords:** oral squamous cell carcinoma, Mannich bases, M2-type pyruvate kinase, autophagy, molecular docking simulation

## Abstract

**Background/Objectives:** Pyruvate kinase M2, a central regulator of cancer cell metabolism, has garnered significant attention as a promising target for disrupting the metabolic adaptability of tumor cells. This study explores the potential of the Mannich base derived from lawsone (**MB-6a**) to interfere with PKM2 enzymatic activity both in vitro and in silico. **Methods:** The antiproliferative potential of **MB-6a** was tested using MTT assay in various cell lines, including SCC-9, Hep-G2, HT-29, B16-F10, and normal human gingival fibroblast (HGF). The inhibition of PKM2 mediated by **MB-6a** was assessed using an LDH-coupled assay and by measuring ATP production. Docking studies and molecular dynamics calculations were performed using Autodock 4 and GROMACS, respectively, on the tetrameric PKM2 crystallographic structure. **Results:** The Mannich base **6a** demonstrated selective cytotoxicity against all cancer cell lines tested without affecting cell migration, with the highest selectivity index (SI) of 4.63 in SCC-9, followed by B16-F10 (SI = 3.9), Hep-G2 (SI = 3.4), and HT-29 (SI = 2.03). The compound effectively inhibited PKM2 glycolytic activity, leading to a reduction of ATP production both in the enzymatic reaction and in cells treated with this naphthoquinone derivative. **MB-6a** showed favorable binding to PKM2 in the ATP-bound monomers through docking studies (PDB ID: 4FXF; binding affinity scores ranging from −6.94 to −9.79 kcal/mol) and MD simulations, revealing binding affinities stabilized by key interactions including hydrogen bonds, halogen bonds, and hydrophobic contacts. **Conclusions:** The findings suggest that **MB-6a** exerts its antiproliferative activity by disrupting cell glucose metabolism, consequently reducing ATP production and triggering energetic collapse in cancer cells. This study highlights the potential of **MB-6a** as a lead compound targeting PKM2 and warrants further investigation into its mechanism of action and potential clinical applications.

## 1. Introduction

Cancer is the name given to a set of more than one hundred diseases characterized by uncontrolled cell growth. It has a monoclonal origin, arising from several factors, including the accumulation of mutations and epigenetic alterations in DNA [1]. According to recent reports, there were more than 20 million new cases of cancer diagnosed and almost 10 million deaths. The four most incident cancers are lung, breast, colorectal, and prostate, respectively, accounting for approximately 7 million cases [2]. It is predicted that by 2040, there will be 28 million cases worldwide, requiring a growing investment in the development of new antineoplastic agents [3]. The most used methods in cancer treatment are surgery, chemotherapy, and radiotherapy. However, some patients experience recurrence or do not respond well to treatment, making it important to detect new targets for the development of more effective treatments [4]. In recent years, the development of new anticancer agents, including synthetic analogs of natural products, has increasingly focused on kinase inhibitors, which are essential for regulating key processes often disrupted in cancer cells [5]. One relevant aspect of cancer that can be used as a target in therapy is the abnormal energy metabolism in cancer cells, considered a cancer marker [1]. In normal cells under aerobic conditions, two molecules of ATP are produced through glycolysis, and the product pyruvate enters the citric acid cycle to produce a greater amount of ATP. In contrast, cancer cells are characterized by having a more intense glycolytic process and a reduced ATP production ratio in mitochondria, resulting in lactate formation even in the presence of oxygen. This phenomenon is known as the Warburg effect, in which the cell compensates for the lower ATP yield from glycolysis compared to the electron transport chain with a more efficient capture of glucose from the environment [6]. Even though this process produces less ATP, it enhances the generation of glycolytic intermediates that can be used by the constantly growing and proliferating cancer cells in the synthesis of biomolecules such as lipids and nucleotides [6]. Furthermore, the lactate produced contributes to the acidification of the tumor microenvironment, which in turn increases inflammation and modulates immune response [7].

A key enzyme in glycolysis is pyruvate kinase M2 (PKM2). This enzyme, in its tetrameric form stabilized by allosteric factor fructose-1,6-bisphosphate (FBP), catalyzes the final step of glycolysis, where a high-energy phosphate group is transferred from phosphoenolpyruvate (PEP) to ADP, forming ATP [8]. Pyruvate kinase has four isoforms in mammals: PKM1, PKM2, PKL, and PKR. PKR is expressed in red blood cells, PKL is expressed in the liver, kidneys, and intestines, while PKM1 is expressed in tissues that require a high energy supply, such as the heart, brain, and muscles [9]. The isoform PKM2 is expressed in proliferating cells, such as embryonic and cancer cells. In addition to participating in the energy metabolism of cancer cells, PKM2 forms dimers that translocate to the nucleus and act as a transcription factor, regulating the expression of various oncogenes involved in cell proliferation and glucose uptake [10,11]. Another important function of PKM2 is its protein kinase activity, as it is described as capable of translocating to the mitochondria, phosphorylating the antiapoptotic protein BCL2, and preventing its degradation and cell death by apoptosis [12]. The overexpression of PKM2 is also associated with multidrug resistance, metastasis, and angiogenesis [13]. The expression of PKM2 in cancer cells makes it an attractive target for substances designed for cancer treatment.

On the other hand, quinones are substances found in plants, fungi, and algae. They are part of various processes, such as the respiratory chain and blood coagulation, among others [14]. Naphthoquinones are quinones that have a naphthalene ring in their structure. These compounds have a wide therapeutic application for diseases such as Chagas disease, fungal infections, antiviral diseases, leishmaniasis, and antineoplastic activity [15,16,17,18]. In a previous study by our group, a screening of 16 Mannich bases derived from 1,4-naphthoquinone was conducted. Among these, substance **6a** (methyl benzyl((4-chlorophenyl)(3-((methoxycarbonyl)oxy)-1,4-dioxo-1,4-dihydronaphthalen-2-yl)methyl)carbamate), now called **MB-6a** (Figure 1), proved to be the most active and selective in oral squamous cell carcinoma (OSCC) lines SCC-9, SCC-4, and SCC-25, as well as in normal human primary fibroblast cells [19].

Furthermore, the substance demonstrated its antitumor potential by inducing autophagy in SCC-9, followed by apoptosis. Molecular docking performed in that study predicted an interaction between **MB-6a** and the PKM2 enzyme, showing a binding pocket in the ATP-binding cavity of PKM2. Knowing that PKM2 is related to energy metabolism and that its inhibition can lead to energy deprivation, the present study aims to further investigate this interaction between this possible ligand to determine whether the interaction between **MB-6a** and PKM2 is responsible for the mode of action of **MB-6a** employing enzymatic assays in vitro and molecular dynamics simulation in silico.

## 2. Materials and Methods

### 2.1. Biological Assays

#### 2.1.1. Cells and Reagents

Tumor cell lines SCC-9 (human tongue OSCC), HT-29 (colorectal adenocarcinoma), Hep-G2 (hepatocellular carcinoma), and B16-F10 (melanoma) were obtained from ATCC (Masassas, VA, USA) (CRL-1629, HTB-38, HB-8065, and CRL-6475, respectively). Normal human primary gingival fibroblasts (HGF) were obtained by ATCC (Masassas, VA, USA) (PCS201 018). The OSCC line was maintained in 1:1 DMEM/F12 (Dulbecco’s Modified Eagle Medium and Ham’s F12 medium; Gibco (Thermo Fisher, Waltham, MA, USA) supplemented with 10% (*v*/*v*) FBS (fetal bovine serum; Invitrogen, ThermoFisher, Waltham, MA, USA) and 400 ng/mL hydrocortisone (Sigma-Aldrich Co., St. Louis, MO, USA). Other cancer cell lines and normal oral human fibroblast were cultured in DMEM supplemented with 10% (*v*/*v*) FBS. All cells were incubated in a humidified environment containing 5% CO_2_ at 37 °C. Naphthoquinone **MB-6a** was synthesized at the Chemistry Institute of the Fluminense Federal University [19]. For in vitro tests, it was solubilized in 100% DMSO (Sigma-Aldrich Co., St. Louis, MO, USA) to a stock concentration of 10 mM. Carboplatin (Fauldcarbo^®^; Libbs Farmacêutica, São Paulo, SP, Brazil) was used as a control.

#### 2.1.2. Cell Viability Assay (Cytotoxicity)

Metabolic activity (cell viability) was measured using the MTT assay according to previous study [20]. Tumor and normal cells were seeded in 96-well plates (5 × 10^3^ cells per well). After 24 h (for cancer cells) and 72 h (for normal cells), the medium was removed, and subsequently, treatment was carried out by applying five different concentrations of **MB-6a** ranging from 0.06 µM to 150 µM and 0.4 µM to 800 µM against cancer cells and normal cells, respectively, and the control ranging from 6.25 µM to 800 µM and 12.5 µM to 1600 µM in cancer cells and fibroblasts, respectively. After 48 h of incubation in the presence of substances, cells were then incubated for 3.5 h with 0.5 mg/mL MTT reagent (3,4,5-dimethiazol-2,5-diphenyltetrazolium bromide) (Sigma-Aldrich Co., St. Louis, MO, USA). Subsequently, formazan crystals were dissolved with MTT solvent solution (methanol/DMSO 1:1 *v*/*v*), and absorbance was measured at 560 nm using an EPOCH microplate spectrophotometer (BioTek Instruments, Winooski, VT, USA), with background absorbance at 670 nm subtracted.

#### 2.1.3. Cell Migration Assay

The inhibitory capacity of cell migration by substance **MB-6a** was assessed following a previously described protocol [21]. For the experiment, 2 × 10^5^ cells from SCC-9 cell line were seeded in a 35 mm Petri dish and incubated at 37 °C with 5% CO_2_ until the formation of a cell monolayer. After this time, the supernatant was removed, cells were washed once with phosphate-buffered saline (PBS), and then fresh medium (2.0 mL) in the presence of Mitomycin C (MMC), 1-amino-9a-methoxy-7-methyl-4,7,9,9a-tetrahydro-3H-furo [3,4:6,7]naphtho [1,2-d]imidazol-2(1H)-one (Sigma-Aldrich Corporation^®^, St. Louis, MO, USA) at a concentration of 0.5 mg/mL, in a volume of 0.5 µL/mL was added to the plate, which was then returned to the incubator for 2 h. At the end of this time, scratch wounds were created in the cell monolayer, and the plate was washed three times with PBS for proper removal of cell debris. Treatment with substances **MB-6a** and DMSO control were added at a sublethal concentration of 1/8 of the IC_50_. The migratory potential of the cells was assessed by means of photographs taken every 12 h of the same fields (scratches) using Leica DMi1 microscope (Leica Microsystems, Wetzlar, Germany). Measurements between the captured margins were collected using ImageJ analysis software v. 1,53e (National Institute of Health, Bethesda, MD, USA). An average of five different distances between the edges was obtained for each time point. At the initial time point of each situation, the distance between the wound edges was set to 100%, representing the open wound. The percentage of wound closure at each subsequent time point is calculated in relation to the initial distance (time point 0 h).

#### 2.1.4. Statistical Analysis

Half maximal inhibitory concentration (IC_50_) values were obtained from the MTT assay of at least three independent replicates through non-linear regression analysis using GraphPad Prism 5.0 software (Intuitive Software for Science, San Diego, CA, USA). Data were presented as means ± standard deviation (SD). A log dose–response curve (inhibitor vs. response) using the least squares method was used to determine the IC_50_ and SD from the data. The selectivity index (SI) was calculated as the ratio between the IC_50_ in normal cells and the IC_50_ in cancer cells, obtained by dividing the IC_50_ of normal cells by the IC_50_ of cancer cells.

### 2.2. PKM2 Activity

#### 2.2.1. Kinetic PKM2 Activity In Vitro

PK activity was measured using LDH-coupled assay according to published protocols [22]. Reactions for each assay contained 7.5 pH buffer containing the following components: recombinant human PKM2 30 ng/reaction (Sigma-Aldrich Co., St. Louis, MO, USA #SAE0021), 50 mM Tris-HCl, 100 mM KCl, 5 mM MgCl_2_, 0.6 mM ADP, 0.5 mM PEP, 180 µM NADH, 10 µM FBP, and 8 units LDH (Sigma-Aldrich Co., St. Louis, MO, USA #10 127 230 001). It is noteworthy that PEP was only added immediately before measurement to start the reaction. The change in absorbance at 340 nm due to oxidation of NADH by LDH was measured every minute for 30 min using an EPOCH microplate spectrophotometer (BioTek Instruments, Winooski, VT, USA). The activity of PK was defined as percentage relative to the indicated control.

#### 2.2.2. Quantification of ATP Production from PKM2 Reaction

The measurement of PKM2 activity through ATP production was conducted to confirm whether the enzymatic inhibition observed in the previous assay was indeed due to PKM2. The master mix was prepared as indicated above. In a microtube containing the master mix prepared in the presence of 1 × IC_50_ (1.039 nm) of **MB-6a**, the control received the addition of 0.5 mM PEP for 3 min. Afterward, CellTiter-Glo^®^ reagent (Promega, Madison, WI, USA) was added as indicated by manufacturer’s instruction, and the samples were kept shielded from light for 2 min. After the specified time, the sample was taken for reading in a TD 2020 luminometer (Turner Designs, Sunnyvale, CA, USA).

#### 2.2.3. Quantification of Intracellular ATP Production

CellTiter-Glo^®^ 2.0 Assay (Promega, Madison, WI, USA) was used to measure intracellular adenosine triphosphate (ATP) concentration associated with violet crystal assay to normalize ATP concentration by cell numbers. SCC-9 cells were seeded in 96-well plates (5 × 10^3^ cells/well), and after 24 h of incubation, cells were treated with compound **MB-6a** in four different concentrations, ranging from 0.1 µM to 100 µM. DMSO was used as negative control to normalize ATP concentration in each treatment. After 12 h, ATP concentration was assessed as instructed by the manufacturer, and cell viability was determined by crystal violet assay.

### 2.3. In Silico Studies

#### 2.3.1. Construction of the PKM2-MB-6a Complex

To elucidate the molecular mechanisms underlying the in vitro effects observed with **MB-6a** treatment, we investigated PKM2 as a potential molecular target through in silico approaches. The **MB-6a** molecule was constructed, and its geometry optimized using the Avogadro2 software package v.1.97.0 [23]. To accurately mimic the experimental conditions at pH 7.5, the compound **MB-6a** protonation state was evaluated using (Marvin JS version 24.3.0, calculation module developed by ChemAxon, https://playground.calculators.cxn.io/, 8 December 2024) [24]. Similarly, the protonation states of PKM2 protein were adjusted to reflect the same pH using the PROPKA3 [25] algorithm through the APBS-PDB2PQR web server (server.poissonboltzmann.org/pdb2pqr) [26] ensuring consistency between the computational model and experimental conditions.

#### 2.3.2. Molecular Docking

Docking was conducted using AutoDock4 v4.2.6 in conjunction with AutoDockTools v.1.5.6 [27], employing the Lamarckian Genetic Algorithm [28]. A total of 200 generations were calculated, with other parameters set to default. The grid box was manually set to encompass the entire ATP structure, its binding pocket, and nearby surface, with 60 × 60 × 60 points (X, Y, Z) with 0.375 Å spacing, centered on coordinates 41.659, 4.204, 61.869 (X, Y, Z). Two-dimensional ligand–protein interaction diagrams were performed using Maestro (Schrödinger, LLC, New York, NY, USA, 2024).

#### 2.3.3. Molecular Dynamics Simulation

To assess the stability of the complex (PKM2 tetramer with one **MB-6a** ligand docked to each of the four chains and four crystallographic bound fructose-1,6-biphosphate (FBP) molecules), a 300 ns molecular dynamics (MD) simulation was performed.

The PKM2 tetrameric complex with **MB-6a** (PKM2-**MB-6a**) system was constructed using the CHARMM-GUI solution builder module [29]. The initial structure for the MD simulation was obtained from prior molecular docking studies by selecting the lowest energy pose from triplicate runs. The simulation included the tetrameric form of PKM2 with one **MB-6a** molecule bound to each monomer binding site, totaling four **MB-6a** molecules in the complex. The complex was solvated in an octahedral box with dimensions ensuring a minimum distance of 1.4 nm between the complex and the box edges. The TIP3P water model [30] was used for solvation. The system was neutralized by adding Na^+^ and Cl^−^ counterions with a final concentration of 0.15 M. All MD simulations were performed using the GROMACS 2024.1 software package [31] with CHARMM36m force field [32] for the protein. Parameters for compound **MB-6a** were generated using the CHARMM General force field (CGenFF) [33,34,35] via the ParamChem service, accessible through CHARMM-GUI. Periodic boundary conditions were applied in all simulations. Non-bonded interactions were calculated with a cut-off distance of 1.0 nm for both van der Waals and short-range electrostatics. The Lennard–Jones potential was smoothly switched off over 0.8 to 1.0 nm by a force-based switching function [36]. Long-range electrostatic interactions were treated using the Particle Mesh Ewald (PME) method [37,38]. To maintain constant bond lengths involving hydrogen atoms, the Linear Constraint Solver (LINCS) algorithm [39] was employed, while the SETTLE algorithm [40] was used specifically for water molecules.

Energy minimization was performed to remove unfavorable interactions between the complex and the solvent. Initially, 5000 steps of the steepest descent method were applied with positional restraints on the backbone and ligand heavy atoms (400 kJ·mol^−1^·nm^−2^) and protein side chain heavy atoms (40 kJ·mol^−1^·nm^−2^). This was followed by an additional 5000 steps of the conjugate gradient method without any positional restraints, allowing the entire system to relax freely. Subsequently, the system was gradually heated from 0 K to 303.15 K over 800 ps and maintained at 303.15 K for an additional 200 ps, totaling 1 ns of equilibration with a time step of 1 fs. Temperature control was achieved using the Berendsen thermostat [41] with a coupling constant of 1 ps under the NVT ensemble, applying the same positional restraints as during initial minimization. Initial velocities were assigned from a Maxwell–Boltzmann distribution corresponding to the starting temperature. Pressure equilibration was performed over 2 ns using an isothermal–isobaric (NPT) ensemble to adjust the solvent density. Positional restraint force constants on the backbone and ligand heavy atoms (initially 400 kJ·mol^−1^·nm^−2^) and protein side chain heavy atoms (initially 40 kJ·mol^−1^·nm^−2^) were gradually decreased every 250 ps over eight cycles until they reached 0 kJ·mol^−1^·nm^−2^. This gradual reduction allowed the system to relax progressively without abrupt changes. Finally, a production MD simulation of 300 ns was conducted without any positional restraints at constant pressure (1.0 bar) controlled by the Parrinello–Rahman barostat [42] with a coupling constant of 5.0 ps, and constant temperature (303.15 K) controlled by the Nosé–Hoover thermostat [43,44] with a coupling constant of 1.0 ps. The integration time step was set to 2 fs.

Due to the σ-hole feature, halogenated compounds like **MB-6a** include a positively charged virtual site at the apex of halogen atoms, which was considered in the simulation parameters.

Conformations were collected every 200 ps, and 50 snapshots for the last 100 ns were extracted for the gas-phase binding energy calculations via the Molecular Mechanics Poisson–Boltzmann Surface Area (MMPBSA) method [45] The resulting trajectories were analyzed using the GROMACS modules [46] and gmx MMPBSA software [47]. Trajectories were visualized using VMD v.1.9.3 [48], and cluster analysis was performed using the gromos method [49], with a cutoff of 2 Å. Images were generated using PyMol (The PyMOL Molecular Graphics System, Version 3.0 Schrödinger, LLC).

#### 2.3.4. Interaction Energy Assessment

The binding interaction energy between PKM2 and **MB-6a** was calculated employing the Molecular Mechanics Poisson–Boltzmann Surface Area (MMPBSA) approach [50] implemented in gmx_MMPBSA v1.6.3 [47]. The binding energy was computed as the difference between the ensemble-averaged energy of the complex and the sum of the energies of the isolated protein and ligand.

The energy components were decomposed into molecular mechanics energy, which encompasses internal, electrostatic, and van der Waals interactions, and solvation energy. The solvation term comprises both polar and nonpolar contributions, where the polar component was determined by solving the Poisson–Boltzmann equation. The nonpolar solvation term was calculated considering repulsive and dispersion components using the Linear Combination of Pairwise Overlaps (LCPO) method [51] with a probe radius of 1.4 Å.

The LCPO calculations employed standard empirical parameters: cavity surface tension (γ) of 0.0378 kcal/mol.Å^−2^ and cavity offset (β) of −0.5692 kcal/mol. The dispersion term was computed using the surface integration approach described by Tan et al. [52]. The ionic strength was set to 0.15 mM. The atomic radii set for Poisson–Boltzmann calculations have been adjusted according to the charmm_radii, implemented in gmx_MMPBSA. The analysis was performed on 50 equidistant snapshots extracted from the last 100 ns (200–300 ns) of the MD trajectory at 2 ns intervals. Images were generated using PyMol (Molecular Graphics System, Version 3.0 Schrödinger, LLC), while 2D diagrams were built using Maestro (Schrödinger, LLC, New York, NY, USA, 2024).

## 3. Results and Discussion

### 3.1. Compound MB-6a Cytotoxicity Is Related to the Interference in PKM2 Glycolytic Activity

#### 3.1.1. Compound MB-6a Is More Active and Selective in OSCC Compared to Other Types of Cancer

In the previous study by Borges and coworkers (2022) [19], the antitumor activities and molecular mechanisms of action of sixteen Mannich bases derived from lawsone were investigated exclusively in oral cancer cell lines. In that study, compound **MB-6a** stood out as the most cytotoxic and selective naphthoquinone at that point. The present study expands on these findings by examining the antiproliferative spectrum of **MB-6a** in various cancer cell lines. Naphthoquinone **MB-6a** was subjected to an MTT cell viability assay in SCC-9 (oral squamous cell carcinoma), HT-29 (colorectal adenocarcinoma), Hep-G2 (hepatocellular carcinoma), B16-F10 (murine melanoma) and normal human gingival fibroblast (HGF). The consistency of the results regarding the antiproliferative potential of naphthoquinone **MB-6a** is evidenced by the similar IC_50_ values in SCC-9 observed in the present study and the previous one, which were 56.74 µM and 56.20 µM, respectively. This confirms the reproducibility of studies on **MB-6a**, highlighting it as a promising molecule for lead development.

For the cell viability assay, carboplatin was the control. Carboplatin is a chemotherapeutic agent used in the treatment of a wide range of cancers, for example, oral cancer [53]. The increasing reports of drug resistance over time for different chemotherapeutic agents, including carboplatin, demand the testing of new substances against various cancers [54].

The presence of **MB-6a** in the cancer cell lines tested inhibited cell viability in a dose-dependent manner. The most sensitive cell line was SCC-9, with the lowest IC_50_ (56.74 µM), followed by B16-F10 (66.42 µM), Hep-G2 (76.69 µM) and HT-29 (129.0 µM). The IC_50_ value for normal cells was significantly higher, indicating the selectivity of **MB-6a** for cancer cells, as indicated in Table 1. The effect of 1,4-naphthoquinone against Hep-G2 cells synthesized by Wang and coworkers (2018) [55] was superior to **MB-6a**, presenting a lower IC_50_ value. However, a high degree of cytotoxicity against hepatocarcinoma cells may be a predictive parameter of hepatotoxicity [56]. In the previously published work, compound **MB-6a** showed a low hepatotoxic profile at tolerated doses in an acute toxicity assay using C57BL/6 mice, which is consistent with the in vitro results obtained in Hep-G2 cells.

One of the most relevant parameters in the development of drug candidates is selectivity. It is an index that can be interpreted as the ratio between the cytotoxicity of a compound towards the target (cancer cell) and its cytotoxicity in normal cells. Selectivity indices higher than 2 indicate that a chemotherapeutic agent or a drug candidate for cancer treatment is selective [18]. Wang and coworkers (2015) [57] tested a series of 1,4-naphthoquinones derivatives against HT-29, SW480, Hep-G2, MCF-7, and HL-60 cell lines using the MTT assay, and the most cytotoxic substance showed suppression of cancer cells survival in a selective manner, despite lacking information related to in vivo toxicity. Similarly, hybrid molecules combining naphthoquinone with acridine have shown strong cytotoxic effects against OSCC cell lines, with favorable in vitro and in vivo outcomes [58].

The selectivity index (SI) values obtained for substance **MB-6a** indicate that it exhibits greater cytotoxicity in cancer cells compared to normal cells, particularly in the SCC-9 line (SI = 4.63). In both B16-F10 (murine melanoma) and Hep-G2 (hepatocellular carcinoma) cell lines, compound **MB-6a** demonstrated selectivity of 3.9 and 3.4, respectively, outperforming carboplatin (SI = 0.99 and SI = 3.03). However, comparisons with other compounds show varying results. In Hep-G2 cells, compound **MB-6a** demonstrated lower selectivity compared to the spirooxindole-O-naphthoquinone-tetrazole-[1,5-a]pyrimidine hybrid (SI = 20.60) in the work by Wu and Liu (2018) [59], which exhibited significantly greater cytotoxicity and selectivity. In B16-F10 cells, **MB-6a**’s IC_50_ value (66.42 µM) was substantially higher than that of naphthoquinone derivatives substituted with dithiocarbamates (IC_50_ = 0.104 µM) presented by Ning and coworkers, 2018 [60], indicating lower cytotoxicity of **MB-6a**. These findings suggest that while **MB-6a** shows promising selectivity, its cytotoxicity varies across different cell lines. Figure 2A presents these results for **MB-6a** in nonlinear regression curves, allowing the comparison of the different patterns of inhibition between the cell lines tested.

Analysis of Figure 2A reveals that substance **MB-6a** demonstrates a consistent pattern of cytotoxicity across various cancer cell lines, reducing cell viability by approximately 50%, despite the presence of distinct IC_50_ values for SCC-9 (IC_50_: 56.74 µM), HT-29 (IC_50_: 129.0 µM), Hep-G2 (IC_50_: 76.69 µM), and B16-F10 (IC_50_: 66.42 µM). In contrast, HGF exhibited greater resistance to compound **MB-6a**, with an IC_50_ value of 262.9 µM. The assessment of cytotoxicity and selectivity of antitumor agents is of paramount importance, as the majority of current chemotherapeutics lack selectivity, demonstrating cytotoxic effects on both cancerous and normal cells. The non-selectivity of drugs, in general, results in significant adverse effects during therapy, limiting clinical use [61].

Naphthoquinone derivatives may exert their biological activity against several targets, such as bacteria or cancer, through interference in redox balance enhancing intracellular reactive oxygen species concentration, causing damage to biomolecules, which can lead to apoptosis, necroptosis, and autophagy in cancer cells [62,63]. The possible mechanism of action of **MB-6a** has already been investigated. Although **MB-6a** did not initially induce apoptosis, its activity suggests it may lead to cell death through autophagy followed by later apoptosis, supporting its potential as a therapeutic agent in cancer treatments. Some of the outcomes described in PKM2 inhibition include an autophagy-like phenotype in oral cancer cells, cell death through apoptosis, and reduction in cell migration, which led to further investigation of PKM2 as a possible target of **MB-6a** [64,65]. In this context, inhibition of cell migration and enzymatic interference of **MB-6a** in the PKM2 pathway is described later.

To investigate cell migration upon **MB-6a** treatment, a wound-healing assay was performed. In this assay, a scratch (wound) is made in the cell monolayer to monitor the movement of cells toward the wound. For this assay, SCC-9 cells were treated with a sublethal dose equal to 1/8 of the IC_50_ (7.03 µM). Over 24 h, the status of the wound was analyzed in the presence of **MB-6a**. The vehicle (DMSO) was applied under the same conditions. It can be observed that both **MB-6a** and the negative control (DMSO) showed similar wound closure Figure 2B,C. There was also no significant difference in the width of the wound at the tested times, indicating that substance **MB-6a**, under these conditions, has no effect on blocking migration in SCC-9 cells.

PKM2 is related to the activation of STAT3 and other target genes involved in epithelial–mesenchymal transition (EMT), favoring the E-cadherin to N-cadherin switch involved in cell migration. Inhibition of PKM2 is consequently related to suppressed invasion and migration in cancer [66,67]. However, the work by Chen et al. (2015) [68] showed that inhibition of PKM2 in hepatocarcinoma was able to impair cell proliferation but not cell migration. In their work, PKM2 inhibition was related to E-cadherin downregulation, followed by N-cadherin upregulation and enhanced EGFR activity, favoring migration of the knockdown cells in the presence of EGF. In assays performed with **MB-6a**, its activity was similar to Chen’s findings, revealing an antiproliferative potential but not an antimigrative phenotype in treated cells.

#### 3.1.2. Substance MB-6a Demonstrated Inhibitory Potential Against PKM2 Through an Enzymatic Assay

Cancer cells are characterized by having a switch in their energy metabolism called the Warburg effect. In this process, cells, even in the presence of oxygen, undergo intense lactic fermentation, converting the pyruvate obtained from glycolysis into lactate. This process prioritizes the synthesis of biomass over the synthesis of ATP in the citric acid cycle and electron transport chain [69,70]. PKM2 is a key enzyme in this process. It regulates the final stage of glycolysis, with the tetrameric form presenting affinity to phosphoenolpyruvate (PEP) and ADP, catalyzing the formation of pyruvate and ATP [6]. PKM2 is overexpressed in several types of cancer, and its less glycolytic potential compared to M1 isoform results in the accumulation of glycolytic intermediates and their capture by biosynthesis pathways [71].

When converted to its dimeric form, PKM2 increases glucose uptake and facilitates the accumulation of amino acids, lipids, and nucleic acids, promoting tumor proliferation and growth [69]. Inhibition of the enzyme would lead to energy deprivation of the neoplastic cell, thus reversing the tumorigenic effect [69,72].

Taking together previous molecular docking results with the PKM2 enzyme and autophagy-like phenotype of cells treated with **MB-6a** demonstrated in Borges et al. (2022) [19], PKM2 became a possible oncogene targeted by **MB-6a**.

The experimental analysis of PKM2 inhibition upon **MB-6a** treatment was conducted through three approaches. The first step was to determine **MB-6a**’s interference in PKM2 glycolytic activity by performing an in vitro biochemical reaction using the LDH-coupled PKM2 activity assay. This assay is important for screening new PKM2 inhibitor candidates, as it allows the indirect measurement of PKM2 activity through the conversion of pyruvate into lactate by LDH-A, using the oxidation of NADH to NAD+ as the readout [73]. For this assay, the potent PKM2 inhibitor, compound Couma. 6e, was used as the positive control [22].

It can be observed that **MB-6a** strongly decreased PK activity in the LDH-coupled assay at low concentrations, with an IC_50_ of 1039 nM (Figure 3A). At the same time, Couma. 6e stood as a more potent inhibitor in these conditions (IC_50_ = 83 nM). Nevertheless, **MB-6a** inhibits the targeted pathway in a satisfactory manner (IC_50_ = 1039 nM or 1.03 µM) compared to phenolic compounds identified as PKM2 inhibitors such as curcumin (IC_50_ = 1.12 µM), resveratrol (3.07 µM) and comparable to silibinin (0.91 µM) [74].

Second, to confirm the direct inhibition of **MB-6a** in PKM2 activity, an assay was performed to measure the production of ATP during the PK reaction after treatment with 1 × IC_50_ of **MB-6a**. As mentioned above, ATP is one of the products formed by PKM2, and measuring its production could demonstrate whether the inhibition is specific in the target enzyme.

As observed in Figure 3B, treatment with the naphthoquinone **MB-6a** and the control suppressed approximately 50% of ATP production, as expected. That information indicates that the ATPase activity of PK was effectively inhibited and that the mode of action of **MB-6a** is specific to PKM2.

Finally, to further investigate the role of **MB-6a** in energy metabolism, we quantified intracellular ATP in SCC-9 cells. As demonstrated in Figure 3C, **MB-6a** acts in a dose-dependent manner to inhibit ATP production, confirming the metabolism-interfering characteristics of this compound. The less pronounced inhibition of ATP production in this assay, requiring higher concentrations of **MB-6a,** can be attributed to the more complex system of ATP production within cells. For instance, glutamine metabolism can generate citric acid cycle intermediates, promoting the synthesis of FADH_2_ and NADH, which function as electron donors for the electron transport chain and ultimately lead to ATP synthesis [75]. Thus, **MB-6a** contributes to only one part of this complex system by inhibiting PEP metabolization and, consequently, ATP production in the glycolytic pathway.

One of the biological processes that may occur in response to ATP depletion is autophagy, a phenomenon in which cells degrade and recycle proteins and organelles to maintain intracellular homeostasis [76,77]. As described previously [19], **MB-6a** induces an autophagy-like process, suggesting that it can be triggered by ATP depletion in treated cells, leading to a collapse in the energetic metabolism and, ultimately, apoptosis.

Moreover, an important application of PKM2 inhibitors lies in the resensitization of cancer cells. Cao and coworkers (2018) [78] demonstrated that suppressing ATP production leads to intracellular accumulation of drugs, possibly by impairing the proper function of the ATP-dependent Pg-P pump. These findings highlight the potential of PKM2 inhibition as a relevant synergistic approach in combination with chemotherapeutic agents.

These experimental results provide valuable insights, but the molecular mechanisms underlying the interaction of **MB-6a** and PKM2 require further elucidation. Consequently, molecular docking assays and molecular dynamics simulations were conducted to investigate the binding interactions of **MB-6a** with the target protein. These computation studies aim to provide more robust predictive information regarding the biological activity of **MB-6a**.

### 3.2. In Silico Analysis of MB-6a-PKM2 Interaction Indicates That PKM2 Is Potentially Targeted by MB-6a

The original crystallographic structure 4FXF contains human PKM2 in its tetrameric form, with two of the monomers each bound to one ATP molecule [79]. System validation was performed by redocking the ATP using chain B of the PKM2 structure (B_PKM2_). The redocking of ATP was successful, obtaining an RMSD of 0.97 Å and a binding energy score of −26.86 kcal/mol. Following validation, compound **MB-6a** was docked onto each of the four chains of PKM2 tetramer (A_PKM2_-**MB-6a**-A, B_PKM2_-**MB-6a**-B, C_PKM2_-**MB-6a**-C, and D_PKM2_-**MB-6a**-D) with the grid adjusted for each chain using the same parameters. Calculations were completed in triplicate to assess the convergence of poses.

Due to induced fit adaptations, these PKM2 monomers exhibit slightly different active site conformations, comprising two regions connected by a hinge structure. Triplicate docking of compound **MB-6a** onto the previously ATP-bound chains (B_PKM2_ and D_PKM2_) resulted in converged best poses with a similar orientation and mean binding energy scores of −9.79 kcal/mol (±0.005) and −9.50 kcal/mol (±0.07), respectively. In contrast, docking onto the two chains not originally bound to ATP (A_PKM2_ and C_PKM2_) yielded heterogeneous best poses with mean binding energy scores of –7.16 kcal/mol (±0.21) and –6.94 (±0.61), respectively, indicating a lack of convergence (Table 2). **MB-6a**-B and **MB-6a**-D exhibited consistent poses across triplicate docking runs. In contrast, **MB-6a**-A did not converge, yielding three independent poses. **MB-6a**-C showed consensus in two out of three runs, with the third pose being considerably different.

In its best-scoring pose (B_PKM2_-**MB-6a**-B), **MB-6a** formed two salt bridges and two hydrogen bonds with three nearby residues, as revealed by interaction mapping. One salt bridge was formed between the deprotonated hydroxyl group of the lawsone moiety and the His78 residue. Lys207 contributed with two simultaneous interactions: one salt bridge formed with the deprotonated hydroxyl and a hydrogen bond with the adjacent carbonyl. The aforementioned carbonyl group also participated in a hydrogen bond with Ser205. Additionally, the aromatic portion of the lawsone moiety formed a π-cation interaction with Lys207. In the initial frame (i.e., the optimal docking pose employed as the starting configuration for molecular dynamics simulations), the chlorobenzene moiety of **MB-6a** was oriented towards the solvent, exhibiting no halogen or other interactions (Figure 4). This observation might be influenced by the lack of detailed σ-hole parameters on Autodock4 calculations, a limitation attributable to its semiempirical nature [27].

Using the molecular docking results as a starting point, a molecular dynamics (MD) simulation of the PKM2 tetramer was performed. Each monomer was bound to the top-scoring pose of **MB-6a** obtained from docking calculations. Ligand conformational analysis based on root mean square deviation (RMSD) revealed remarkable conformational stability of **MB-6a** across all complexes, with fluctuations generally within 2.0 Å (Figure 5A). This stability is further supported by the cluster analysis performed using the gromos algorithm implemented in the GROMACS package. The analysis used a 2.0 Å cutoff and considered only non-hydrogen atoms of the **MB-6a** molecules. While **MB-6a**-B, **MB-6a**-C, and **MB-6a**-D exhibited up to three distinct clusters, their most representative cluster of conformations dominated the simulation time (99.99%, 88.00%, and 75.61%, respectively). This indicated stable binding modes with minimal conformational variations. The structural overlays in Figure 5B visually demonstrate this conformational consistency throughout the trajectory.

Molecular dynamics analysis was followed by a cluster analysis considering both the ligand and pocket region of the complexes A_PKM2_-**MB-6a**-A, B_PKM2_-**MB-6a**-B, and D_PKM2_-**MB-6a**-D. The complex formed by CPKM2-**MB-6a**-C quickly dissociated and was therefore not considered further.

**MB-6a**-A exhibited alternating cluster poses, with each pose lasting approximately 30 ns to 180 ns. At this point, it predominantly adopted the major cluster pose with minor perturbations until the end of the simulation, totaling approximately 115 ns. Through the simulation, **MB-6a**-A remained anchored in the A_PKM2_ pocket via its chlorobenzene region, while other regions dynamically interacted with nearby residues. The molecule also exhibited rotational motion around its axis, with the lawsone group interacting with either the larger or smaller PKM2 domains connected via the hinge region (Figure 6A).

Throughout the simulation, **MB-6a**-B maintained two predominant poses. At approximately 25 ns, **MB-6a**-B adopted the most prevalent pose, which it maintained for about 127 ns with minor perturbations. In this pose, **MB-6a**-B was anchored to B_PKM2_ through its chlorobenzene group, which pointed more towards the smaller domain than into the pocket itself, unlike **MB-6a**-A. At around 180 ns, the hinge region underwent a slight opening, leading to an adjustment of **MB-6a**-B, which then adopted the second most prevalent pose. This pose was visually more similar to that of **MB-6a**-A, with the chlorobenzene group pointing directly into the B_PKM2_ pocket cavity. This pose was maintained for the remaining 120 ns of the simulation (Figure 6B).

**MB-6a**-D initially exhibited a quick dissociation from the pocket, interacting with adjacent residues until around 50 ns, at which point it returned to the pocket. Similar to other **MB-6a** molecules, **MB-6a**-D was anchored through its chlorobenzene group while other groups interacted with nearby D_PKM2_ residues. At around 56 ns, **MB-6a**-D stabilized in the second most prevalent cluster pose, in which the lawsone group interacted with the smaller chain D_PKM2_ domain. However, the aromatic ring of **MB-6a**-D did not significantly interact with nearby residues, leading to varied poses. This cluster pose was interrupted at 99 ns but reappeared at around 150 ns and persisted until 180 ns. Despite remaining anchored through the chlorobenzene in the D_PKM2_ pocket, constant rotation around its axis promoted a hiatus between clusters due to the instability of other **MB-6a**-D groups, which frequently varied their poses. At 228 ns, the lawsone group regained its interaction with the smaller D_PKM2_ domain, clustering into the most prevalent pose, which remained largely unchanged with minor perturbations until the end of the simulation at 300 ns, totaling approximately 72 ns (Figure 6C).

To evaluate the binding affinity and characterize the molecular interactions between **MB-6a** and PKM2, we performed Molecular Mechanics Poisson–Boltzmann Surface Area (MMPBSA) calculations on the last 100 ns (200–300 ns) of the MD trajectory. This time frame represents the clusters that were maintained until the end of the simulation. The MMPBSA calculations revealed differences in binding energies across the PKM2 chains (Table 3). Analysis of the energetic components demonstrated favorable binding between three of the four PKM2 chains and their respective **MB-6a** molecules: B_PKM2_-**MB-6a**-B exhibited the strongest affinity with a binding energy (ΔG_bind_) of −28.88 ± 8.58 kcal/mol), followed by APKM2-**MB-6a**-A with ΔG_bind_ = −26.71 ± 7.29 kcal/mol), and D_PKM2_-**MB-6a**-D with ΔG_bind_ = −22.36 ± 5.20 kcal/mol). **MB-6a**-C dissociated from C_PKM2_ during the MD simulation, precluding MMPBSA calculations for this chain and indicating potential binding site instability in this region.

The binding energy decomposition revealed multiple contributing factors. Van der Waals interactions showed significant contributions, particularly in B_PKM2_-**MB-6a**-B (−43.60 ± 5.52 kcal/mol). The A_PKM2_-**MB-6a**-A and D_PKM2_-**MB-6a**-D complexes exhibited moderate van der Waals contributions (−28.34 ± 4.33 kcal/mol and −32.78 ± 5.08 kcal/mol, respectively). Strong favorable electrostatic interactions were observed across all three complexes, with B_PKM2_-**MB-6a**-B showing the strongest contribution (−157.56 ± 18.43 kcal/mol), closely followed by A_PKM2_-**MB-6a**-A (−153.08 ± 30.42 kcal/mol), and chain D_PKM2_-**MB-6a**-D (−133.88 ± 17.64 kcal/mol). These favorable electrostatic interactions were partially compensated by polar solvation penalties, which were highest in the B_PKM2_-**MB-6a**-B complex (177.60 ± 15.85 kcal/mol) and marginally lower in A_PKM2_-**MB-6a**-A and D_PKM2_-**MB-6a**-D complexes (158.55 ± 27.18 and 147.86 ± 16.48 kcal/mol, respectively). The non-polar solvation energy provided additional favorable contributions to the binding, particularly in the B_PKM2_-**MB-6a**-B complex (−5.31 ± 0.13 kcal/mol). The greater stability of the B_PKM2_-**MB-6a**-B complex can be primarily attributed to more favorable van der Waals interactions and slightly stronger electrostatic interactions. Interestingly, although the B_PKM2_-**MB-6a**-B complex also shows the highest polar desolvation penalty, the favorable contributions are sufficient to produce the most negative binding energy among the three chains.

Per-residue energy decomposition identified key interactions contributing to binding stability. Analysis of the binding energy evolution over the final 100 ns revealed dynamic but stable PKM2-**MB-6a** interactions in A_PKM2_-**MB-6a**-A, B_PKM2_-**MB-6a**-B, and D_PKM2_-**MB-6a**-D complexes (Figure 7A). In the A_PKM2_-**MB-6a**-A complex (Figure 7B), several residues show significant favorable contributions to the binding, including charged and polar amino acids that form specific interactions with **MB-6a**. The B_PKM2_-**MB-6a**-B complex’s energy profile (Figure 7C) reveals particularly strong contributions from certain residues, which explains its more favorable total binding energy compared to the other chains. The temporal fluctuations observed are consistent with expected protein dynamics while maintaining favorable interaction energies throughout the analyzed trajectory.

The comprehensive energy analysis across B_PKM2_-**MB-6a**-B and D_PKM2_-**MB-6a**-D complexes demonstrates that, while the overall binding mechanism is conserved, subtle differences in residue contributions lead to variations in binding affinity. This information, combined with structural analysis, could provide a better understanding of the molecular basis for **MB-6a** binding to PKM2.

The structural analysis from the final frame of the MD simulation revealed a complex network of interactions between **MB-6a** and PKM2 (Figure 8). **MB-6a** establishes multiple types of contacts, including charged interactions with Glu residues (negative) and Lys/Arg residues (positive), polar interactions through Ser and Thr residues, and hydrophobic contacts involving Phe and Leu residues. Especially, the lawsone groups of **MB-6a** engage in hydrophobic contacts, while its hydroxyl groups participate in hydrogen bonding networks. These structural features support the energetic analysis and explain the favorable binding energies observed.

An interesting feature observed during structural analysis was the presence of halogen bonds involving the chlorine atom of **MB-6a** in two of the complexes (A_PKM2_-**MB-6a**-A and D_PKM2_-**MB-6a**-D). These interactions were accounted for during the molecular dynamics (MD) simulation by implementing virtual sites for the ligand, enabling a more accurate representation of halogen bonding. The presence of these halogen bonds likely enhances the overall stability and molecular recognition of the protein–ligand complex, complementing other binding interactions observed in the system. It is noteworthy that the negative energy contribution of Glu118, as observed in the per-residue energy decomposition of the A_PKM2_-**MB-6a**-A complex, might be overestimated. This is because the virtual sites used to represent halogen bonds during the simulation were not considered in the MM/PBSA calculations, potentially affecting the energy decomposition analysis of residues involved in these specific interactions.

The MM/PBSA analysis revealed notable binding dynamics between **MB-6a** and PKM2, with favorable binding energies observed in three of the four chains of tetramer (ΔG ranging from −22.36 to −28.88 kcal/mol). Although the B_PKM2_-**MB-6a**-B complex demonstrated the strongest interaction affinity, the overlapping standard deviations suggest comparable binding across these three chains.

While the B_PKM2_-**MB-6a**-B complex exhibits the lowest binding energy, all three analyzed chains demonstrate comparable binding affinities when considering their standard deviations. The dissociation of **MB-6a**-C from C_PKM2_ highlights the complexity of PKM2-**MB-6a** interactions and suggests potential binding site specificity. These findings provide valuable insights for future drug design efforts targeting PKM2, although further investigation may be warranted to understand the factors contributing to C_PKM2_-**MB-6a**-C dissociation.

## 4. Conclusions

Experimental tests using substance **MB-6a** revealed significant antiproliferative effects on SCC-9, B16-F10, and Hep-G2 cells, coupled with reduced toxicity in normal cells. The selectivity index, ranging from 2.03 to 4.63, represents a key characteristic of a drug candidate. The investigation of **MB-6a**’s inhibitory potential on PKM2 glycolytic activity revealed its capacity to disrupt the biochemical reaction mediated by PKM2, ultimately impairing ATP production. This positions **MB-6a** as a promising molecule with the potential to target a critical pathway, especially for cancer cell glycolysis. In silico studies, supported by the experimental results, suggest that **MB-6a** exhibits potential inhibitory effects over PKM2 activity. By interacting with PKM2, **MB-6a** interferes with ATP production, which may subsequently induce autophagic cell death, as previously reported. The presence of halogen bonds in two complexes (A_PKM2_-**MB-6a**-A and D_PKM2_-**MB-6a**-D) and the potential significance of Glu118 in molecular recognition, despite possible overestimation in MMPBSA calculations due to virtual site limitations suggest specific structural features crucial for complex stability. These findings offer valuable insights for future drug design efforts targeting PKM2, demonstrating the prospect for PKM2 inhibitors to selectively act on cancer cells. However, further investigation is warranted to better elucidate the molecular determinants of binding specificity and stability.

## Figures and Tables

**Figure 1 biomedicines-12-02916-f001:**
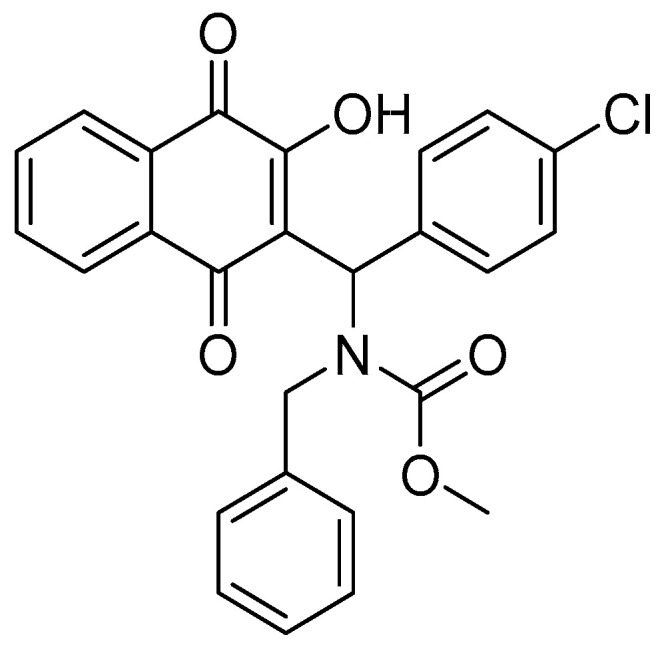
Chemical structure of Mannich base derived from lawsone **MB-6a**.

**Figure 2 biomedicines-12-02916-f002:**
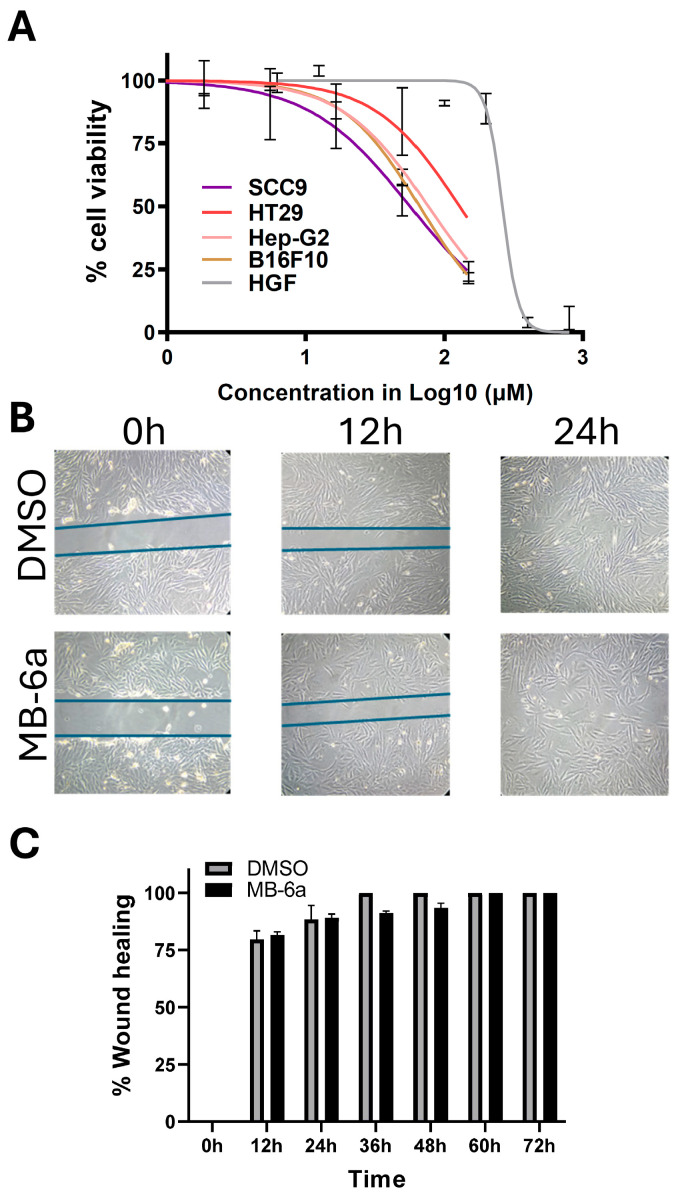
Substance **MB-6a** inhibited cell proliferation but did not impair migration at a sublethal concentration. (**A**) MTT assay results with **MB-6a**. Nonlinear regression curves representing cell viability reduction induced by substance **MB-6a** in SCC-9, HT-29, Hep-G2, B16-F10, and HGF cell lines. The graph represents the curve generated by the number of cells vs. concentration (log 10 µM). (**B**) Cell migration assay using SCC-9 cells. Images represent the scratch (wound) from 0 to 24 h in non-treated cells (DMSO) and treated with a sublethal concentration (7.03 µM) of **MB-6a**. (**C**) Wound width throughout the time. The percentage of wound width after treatment with **MB-6a** and the control (DMSO) at different time points are represented as mean ± SEM. Results were calculated from at least three independent experiments.

**Figure 3 biomedicines-12-02916-f003:**
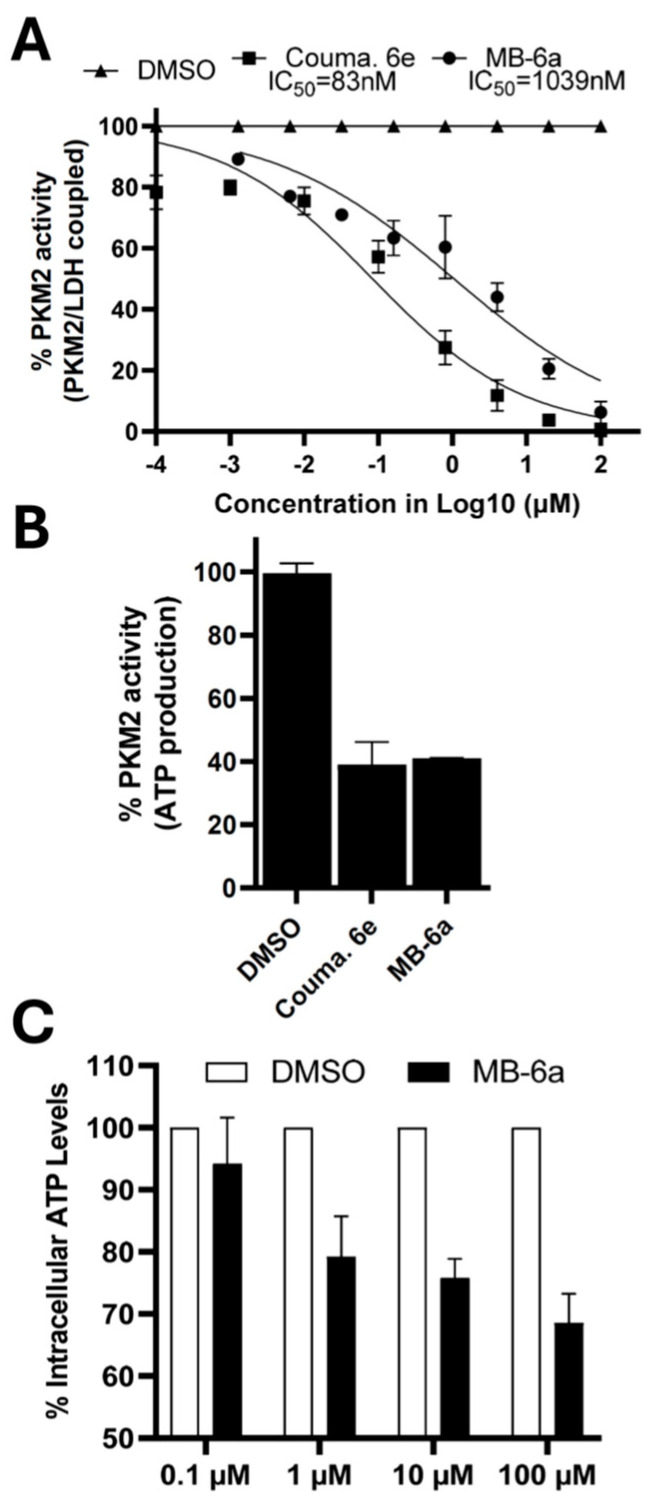
PKM2 activity upon **MB-6a** treatment. (**A**) Glycolytic activity of PKM2 was inhibited by **MB-6a**. LDH coupled assay indicated dose-dependent inhibition of PKM2 activity. Nonlinear regression curves show PKM2 activity after treatment with **MB-6a** at different concentrations (squares). The control, Couma. 6e, is represented by circles, while negative control DMSO is represented by triangles. (**B**) Naphthoquinone **MB-6a** suppressed ATPase activity of PKM2. The degree of inhibition in ATP production by **MB-6a** or the control at a concentration equal to 1 × IC_50_ is depicted. (**C**) Production of ATP in SCC-9 is reduced by **MB-6a**. Intracellular ATP levels were measured after treatment with four different concentrations of **MB-6a**. DMSO was used as the negative control for all assays. Results represent mean ± SEM from at least three independent experiments.

**Figure 4 biomedicines-12-02916-f004:**
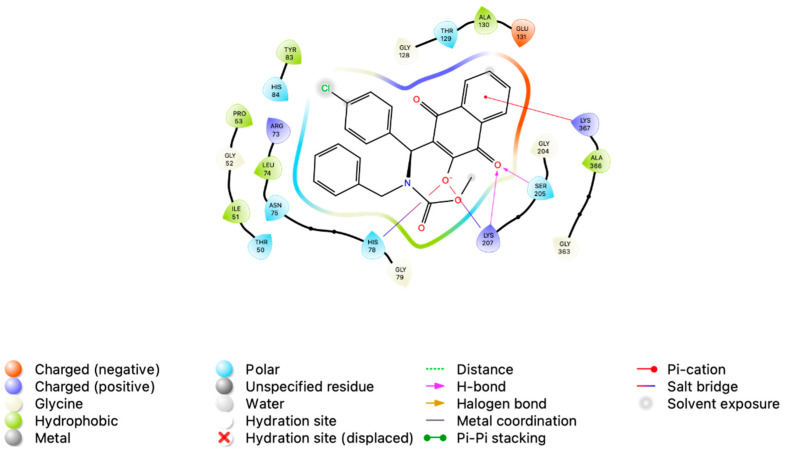
Two-dimensional interaction diagram of **MB-6a** best-scoring pose of **MB-6a** with PKM2. The diagram represents the chain B_PKM2_-**MB-6a**-B complex, with key residues labeled and colored based on their type: negatively charged (orange), positively charged (blue), polar (light blue), and hydrophobic (light green). Two hydrogen bonds (indicated by purple arrows) and two salt bridges (red/blue gradient lines) were formed between the lawsone moiety and nearby residues. Additionally, a cation-π interaction (red line) was formed. The chlorobenzene region was solvent-exposed, as indicated by the gray circle.

**Figure 5 biomedicines-12-02916-f005:**
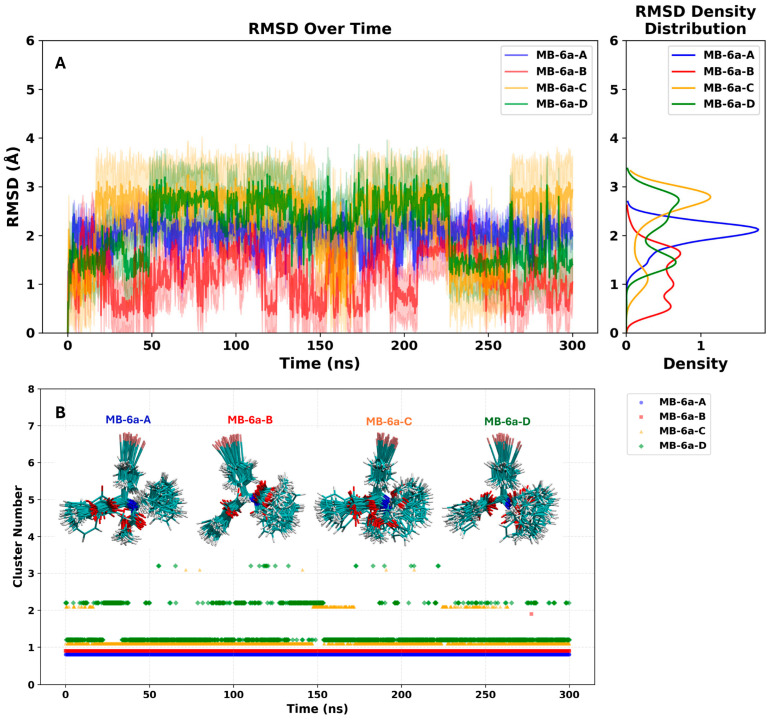
Conformational analysis of **MB-6a** binding to PKM2. (**A**) RMSD evolution over 300 ns of MD simulation for each **MB-6a** molecule (**MB-6a**-A to **MB-6a**-D), showing structural stability within 2Å deviation. The right panel shows the RMSD density distribution for each complex, highlighting the consistency of the structural stability. (**B**) Cluster analysis results with structural superposition of **MB-6a** conformations throughout the MD simulation. Numbers indicate distinct conformational clusters, with the predominant cluster representing 100%, 99.99%, 88.00%, and 75.61% of the simulation time for **MB-6a**-A, **MB-6a**-B, **MB-6a**-C, and **MB-6a**-D, respectively. The high percentage of the predominant cluster for each **MB-6a** molecule indicates stable binding modes with minimal conformational variations.

**Figure 6 biomedicines-12-02916-f006:**
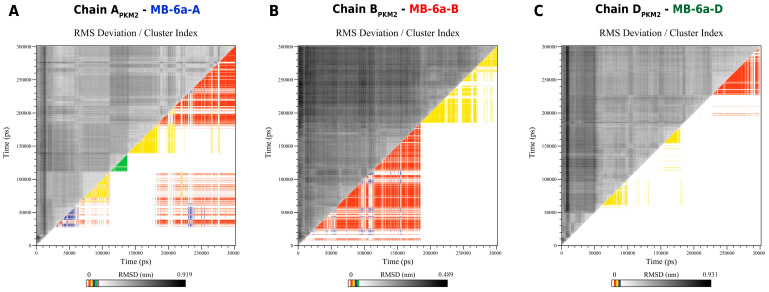
Clustering results of the molecular dynamics simulation. Both axes represent simulation time in ps. The clustering analysis compares simulation frames, grouping similar poses of the ligand pocket region. Cluster analysis was performed for complexes A_PKM2_-**MB-6a**-A (**A**), B_PKM2_-**MB-6a**-B (**B**), and D_PKM2_-**MB-6a**-D (**C**), considering a 2 Å binning using the gromos algorithm. The graphs represent the root mean square deviation (RMSD) matrix on the upper left, and respective clusters plotted throughout 300 ns on the bottom right. Cluster indices are indicated by different colors.

**Figure 7 biomedicines-12-02916-f007:**
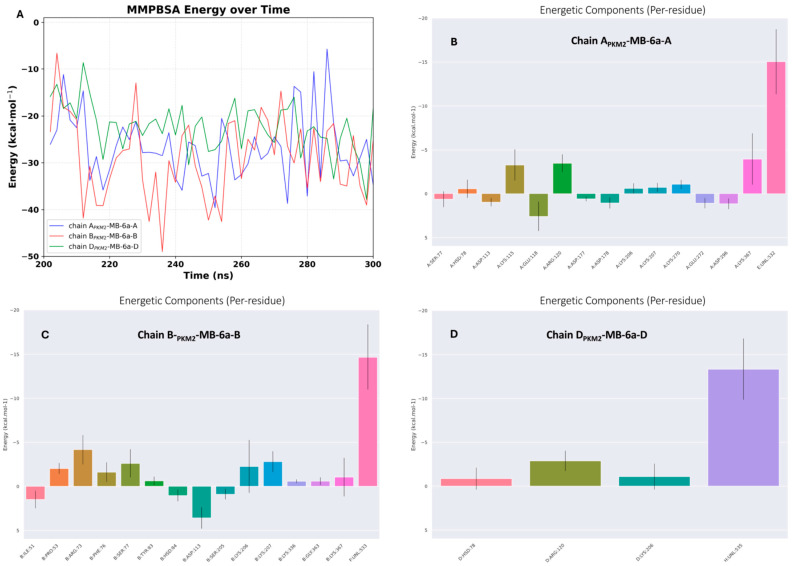
Energetic analysis of PKM2-**MB-6a** binding. (**A**) Time evolution of MMPBSA energy (kcal/mol) during the last 100 ns of MD simulation. (**B**–**D**) Per-residue energy decomposition analysis showing the contribution of individual residues to the total binding energy for A_PKM2_-**MB-6a**-A (**B**), B_PKM2_-**MB-6a**-B (**C**), and D_PKM2_-**MB-6a**-D (**D**). Negative values indicate favorable contributions to binding, while positive values represent unfavorable contributions.

**Figure 8 biomedicines-12-02916-f008:**
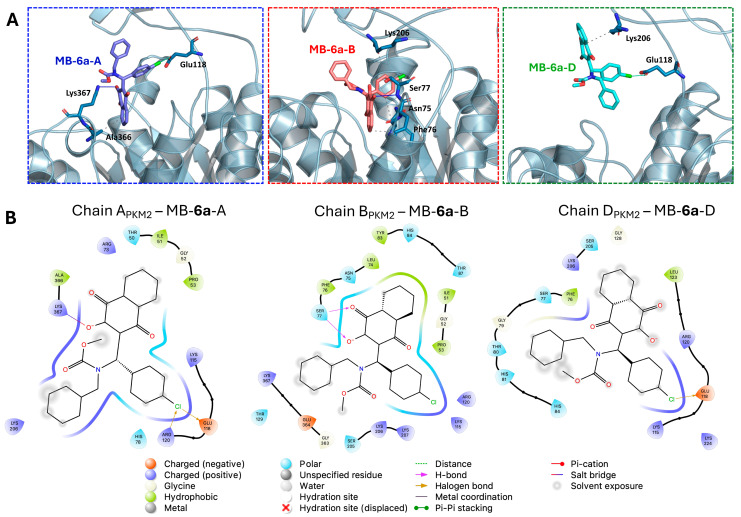
The 2D and 3D interaction diagrams of PKM2-**MB-6a** complexes from the last frame of MD simulation. (**A**) Three-dimensional representation showing the binding mode of **MB-6a** compounds in different chains of PKM2. Hydrogen bonds are highlighted with blue lines, halogen bonds with green lines, and hydrophobic contacts with dashed lines. Key interacting residues are labeled and shown as sticks. The interactions are highlighted in different boxes: **MB-6a**-A interacting with A_PKM2_ (blue box), **MB-6a**-B interacting with B_PKM2_ (red box), and **MB-6a**-D interacting with D_PKM2_ (green box). (**B**) Two-dimensional interaction diagrams showing the binding mode of **MB-6a** with chains A, B, and D of PKM2. Residues are colored according to their type: negatively charged (orange), positively charged (blue), polar (light blue), and hydrophobic (light green). Different types of interactions are represented by distinct line styles: hydrogen bonds (blue arrows), halogen bonds (red arrows), and π-π stacking (green lines). Grey circles indicate solvent exposure. The **MB-6a** structure is shown in the center of each diagram, with key interaction features highlighted.

**Table 1 biomedicines-12-02916-t001:** Compound **MB-6a** exhibited selective cytotoxicity against cancer cells. Determination of IC_50_ (µM), standard deviation, and selectivity index of compounds **MB-6a** and carboplatin in SCC-9, HT-29, Hep-G2, B16-F10, and HGF. The cells were treated with the indicated compounds for 48 h and cell viability was determined by MTT assay. Results represent mean ± SD from at least 3 independent experiments. SD = standard deviation. SI = selectivity index.

Compound	Cancer Cells												Normal Cell	
	SCC-9			Hep-G2			HT-29			B16-F10			HGF	
	IC_50_ (µM)	SD	SI	IC_50_ (µM)	SD	SI	IC_50_ (µM)	SD	SI	IC_50_ (µM)	SD	SI	IC_50_ (µM)	SD
**MB-6a**	56.74	±0.11	4.63	76.69	±0.51	3.4	129.0	±0.05	2.03	66.42	±0.03	3.9	262.9	±0.04
**Carboplatin**	265.3	±0.06	1.69	86.62	±0.02	3.03	174.1	±0.03	1.5	263.1	±0.02	0.99	448.8	±0.06

**Table 2 biomedicines-12-02916-t002:** Docking score results of **MB-6a** on PKM2. Triplicate docking of **MB-6a** to each PKM2 monomer revealed converging poses only on chains that were originally attached to ATP, namely, B_PKM2_ and D_PKM2_.

Complex	Individual Score (kcal/mol)	Average Score (kcal/mol)	SD (kcal/mol)
A_PKM2_-**MB-6a**-A	−7.4	−7.16	±0.21
	−7.13		
	−6.97		
B_PKM2_-**MB-6a**-B	−9.79	−9.79	±0.005
	−9.79		
	−9.8		
C_PKM2_-**MB-6a**-C	−7.62	−6.94	±0.61
	−6.43		
	−6.78		
D_PKM2_-**MB-6a**-D	−9.47	−9.50	±0.07
	−9.46		
	−9.59		

**Table 3 biomedicines-12-02916-t003:** Binding energy components and total binding energy (ΔG_bind_ Total) for the PKM2-**MB-6a** complex across different chains.

Complex	ΔVDWAALS (kcal/mol)	ΔEEL (kcal/mol)	ΔEPB (kcal/mol)	ΔENPOLAR (kcal/mol)	ΔG_bind_ Total (kcal/mol)
A_PKM2_-**MB-6a**-A	−28.34 ± 4.33	−153.08 ± 30.42	158.55 ± 27.18	−3.84 ± 0.51	−26.71 ± 7.29
B_PKM2_-**MB-6a**-B	−43.60 ± 5.52	−157.56 ± 18.43	177.60 ± 15.85	−5.31 ± 0.13	−28.88 ± 8.58
C_PKM2_-**MB-6a**-C	NA	NA	NA	NA	NA
D_PKM2_-**MB-6a**-D	−32.78 ± 5.08	−133.88 ± 17.64	147.86 ± 16.48	−3.56 ± 0.36	−22.36 ± 5.20

Energy values are reported in kcal/mol with their respective standard deviations. Components include van der Waals (ΔVDWAALS), electrostatic (ΔEEL), polar solvation (ΔEPB), and non-polar solvation (ΔENPOLAR) energies. NA indicates data not available for CPKM2-**MB-6a**-C.

## Data Availability

The raw data supporting the conclusions of this article will be made available by the authors on request.

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
