# Peer review of "Mannich Base Derived from Lawsone Inhibits PKM2 and Induces Neoplastic Cell Death"

_biomedicines, 2024, doi:10.3390/biomedicines12122916_

Round 1
Reviewer 1 Report
Comments and Suggestions for Authors
The work titled "Mannich Base Derived from Lawsone inhibits PKM2 and induces neoplastic cell death" represents a significant advancement in the field of cancer therapy.
In this work the authors focused on the Pyruvate kinase M2 as one of the anticancer agents target, they were investigated the potential of the Mannich base derivatives like compound 6a in interfering with PKM2 enzymatic activity both in vitro and in silico. Various cancer cell lines were used in this work besides normal cell lines which considered very important to compare and find the selectivity index. The authors were evaluated the inhibition of PKM2 mediated by 6a compound which was showed selectivity against all cancer cell lines tested without affecting cell migration, with the highest selectivity index (SI) of 4.63 in SCC-9 as well as this compound effectively inhibited PKM2 glycolytic activity, leading to a reduction of ATP production both in the enzymatic reaction and in cells treated with this naphthoquinone derivative. Additionally, the authors were conducted a molecular docking analysis and molecular dynamic simulations. All of these findings seem valuable and suitable contribution to be published in the Biomedicines Journal after justifying the following points:
· It is recommended to summarize the abstract more and give the readers an attractive starting sentence.
· Since just one molecule was evaluated in this work I prefer to use other code rather than 6a because when we see code like 6a as readers we think that there are other molecules like 6b and 6c ….etc
· The cancer cell HEPG2 usually written like Hep-G2, so you have to edit that in the whole manuscript as well as I think there are a code for the used normal cell it is better to write it like the cancer cell lines
· It is recommended to abstract to add the binding affinity score from the docking studies with the PDB code, as well as what do you mean with halogen bonds ?
· The sections’ headings in the abstract should be in the bold style
· In the introduction it is better to add some paragraph regarding the protein kinase last approved drug and you can use the following recent work “Cells 2024, 13, 1656.” Which could improve the story well
· It is recommended to add a figure regarding your molecule 6a to the introduction section
· Line 129 you were wrote different concentrations were used, you have to add these concentrations to the method section too
· 2.1.3 section lines 136-143 this section should be moved to a statistical section, as well as you have to write how did you calculate the SI like “we divide the IC50 of normal cell lines on the IC50 of cancer cell lines”
· In each section of the methods, you have to add references since the method is not novel
· The legend of the table 1 should be edited it is not clear it is a title or footnote, as well as I could not find the unit of the IC50, it is in uM or nM ?? it is better and clearer to write symbol ± for the SD values
· In figure 1 the resolution very bad, as well as correct the concentration word, as well as no need to make the Y axes reach 200% since the cell viability should not be more than 100%
· The conclusion section should be improved, and should not be contain any reference
Best wishes
Author Response
Thank you for all yor comments. They will improve the quality of our work. Below follows the point-by-point rebbutal.
- It is recommended to summarize the abstract more and give the readers an attractive starting sentence.
Thank you for the suggestions. It was applied to the manuscript.
- Since just one molecule was evaluated in this work I prefer to use other code rather than 6a because when we see code like 6a as readers we think that there are other molecules like 6b and 6c ….etc.
Thank you for the suggestion. We updated the molecule code to “MB-6a” (Mannich Base-6a) to maintain the original reference and create a more palatable name. The name was explained in the main text.
- The cancer cell HEPG2 usually written like Hep-G2, so you have to edit that in the whole manuscript as well as I think there are a code for the used normal cell it is better to write it like the cancer cell lines
The HEPG2 code was substituted by Hep-G2 in the main text as requested. ATCC presents the normal cell used in the study as “Primary Gingival Fibroblast; Normal, Human, Adult (HGF) (PCS-201-018)”, hence, we accepted the suggestion and used “HGF” in the main text as the code for the normal cell.
- It is recommended to abstract to add the binding affinity score from the docking studies with the PDB code, as well as what do you mean with halogen bonds?
We thank the reviewer for this important observation. In our study, the halogen bonds were explicitly modeled using the CHARMM36m force field approach, which implements massless virtual "lone pair" (LP) sites on halogen atoms to represent the σ-hole - a region of positive electrostatic potential that develops along the C-X bond axis. This treatment allows for a more accurate representation of the anisotropic charge distribution around halogen atoms, which is important for modeling these directional non-covalent interactions. In our system, this implementation enabled proper modeling of the interactions between the chlorine atoms of compound 6a and the electron-rich region of Glu118 in PKM2. The use of LP sites in our MD simulations provided a more realistic description of these specific protein-ligand interactions, as validated by their stability throughout the trajectories and visualization of the interaction diagrams. Moreover, binding affinity results and PDB code were added to the abstract.
- The sections’ headings in the abstract should be in the bold style.
Corrected in the manuscript.
- In the introduction it is better to add some paragraph regarding the protein kinase last approved drug and you can use the following recent work “Cells 2024, 13, 1656.” Which could improve the story well.
Thank your for the suggestion. It was included as a sentence at the end of the first paragraph.
- It is recommended to add a figure regarding your molecule 6a to the introduction section.
The figure representing the chemical structure of MB|-6a was included as requested.
- Line 129 you were wrote different concentrations were used, you have to add these concentrations to the method section too.
This information was added to the method section 2.1.2.
- 1.3 section lines 136-143 this section should be moved to a statistical section, as well as you have to write how did you calculate the SI like “we divide the IC50 of normal cell lines on the IC50 of cancer cell lines”.
Section 2.1.3 was moved to section 2.1.4 included as a statistical analysis section as suggested. The calculation of SI was better explained.
- In each section of the methods, you have to add references since the method is not novel.
Lacking references of methods were added to the manuscript when necessary in sections “2.1.2 Cell Viability Assay (Cytotoxicity)” and “2.1.3 Cell Migration Assay”.
- The legend of the table 1 should be edited it is not clear it is a title or footnote, as well as I could not find the unit of the IC50, it is in uM or nM ?? it is better and clearer to write symbol ± for the SD values.
Unit of IC50 (“µM”) was included to the legend of table 1 as well to the table itself. The symbol ± was added to the columns of SD.
- In figure 1 the resolution very bad, as well as correct the concentration word, as well as no need to make the Y axes reach 200% since the cell viability should not be more than 100%.
The figure was properly corrected.
- The conclusion section should be improved, and should not be contain any reference.
Conclusion was updated. Alterations emphasize the selective action of MB-6a on cancer cells linking this selective cytotoxicity to PKM2 inhibition.
Reviewer 2 Report
Comments and Suggestions for Authors
In this study, Rubini-Dias et al. demonstrated the cytotoxicity and underlying mechanism of Mannich base 6a, which was previously derived from 1,4-naphthoquinone by the authors. The 6a showed comparable cytotoxicity and selectivity to the similar compounds derived by others in various cancer and normal cell lines. The authors have previously identified that the 6a induced autophagy and late apoptosis in cancer cells. In this study, by using in vitro and in silico approaches, they confirmed the cell death is due to the binding of 6a and PKM2, which leads to the inhibition of ATP production in cancer cells. These results of the 6a suggested its potential as a therapeutic agent in cancer treatment.
Minor issues:
1. Please present the Figure 1A in English.
2. For the wound healing assay in Figure 1B, please show the figure representing data of the wound width at different time points with SEM.
3. Please show the curve of negative control in Figure 2A.
Author Response
Thank you for all yor comments. They will improve the quality of our work. Below follows the point-by-point rebbutal.
Minor reviews
- Please present the Figure 1A in English.
The figure graphically representing the cancer cells viability upon 6a treatment was corrected. “Figure 2a”.
- For the wound healing assay in Figure 1B, please show the figure representing data of the wound width at different time points with SEM.
The figure was updated as requested. Results from treatment and control are now represented with mean with SEM at all different time points tested. Information was also added to the figure’s legend (“Figure 2c”).
- Please show the curve of negative control in Figure 2A.
Curve of negative control (DMSO) was included in this information as well was added to the figure’s legend (Figure 3a). For this experiment, all conditions were normalized by DMSO, thus DMSO is 100% at all concentrations.
Round 2
Reviewer 1 Report
Comments and Suggestions for Authors
The authors were improved the manuscript well regarding each point